# Elevated CO_2_ Impacts on Plant–Pollinator Interactions: A Systematic Review and Free Air Carbon Enrichment Field Study

**DOI:** 10.3390/insects12060512

**Published:** 2021-06-01

**Authors:** Liam M. Crowley, Jonathan P. Sadler, Jeremy Pritchard, Scott A. L. Hayward

**Affiliations:** 1School of Biosciences, The University of Birmingham, Edgbaston, Birmingham B15 2TT, UK; j.pritchard@bham.ac.uk; 2The Birmingham Institute of Forest Research, The University of Birmingham, Edgbaston, Birmingham B15 2TT, UK; j.p.sadler@bham.ac.uk; 3School of Geography, Earth and Environmental Sciences, The University of Birmingham, Edgbaston, Birmingham B15 2TT, UK

**Keywords:** bluebells, bees, hoverflies, phenology, pollination, climate change

## Abstract

**Simple Summary:**

Climate change is having a profound impact on pollination systems, yet we still do not know to what extent increasing concentrations of carbon dioxide (CO_2_) will directly affect the interactions between plants and their pollinators. We review all the existing published literature on the effect of elevated CO_2_ (eCO_2_) on flowering time, nectar and pollen production and plant–pollinator interactions. We also conduct a field experiment to test the effect of eCO_2_ on bluebells and their pollinators. We found that few studies have assessed the impact of eCO_2_ on pollination, and our field data found that bluebells flowered on average 6 days earlier under eCO_2_ conditions. Hoverflies and bumble bees were the main visitors to bluebell flowers, but insect activity was low early in the flowing period. Although we did not find a difference in the number of visits made by insects to bluebell flowers under eCO_2_, or the amount of seeds those flowers produced, the change in the timing of flowering could mean that a mismatch could develop between bluebells and their pollinators in the future, which would affect pollination success.

**Abstract:**

The impact of elevated CO_2_ (eCO_2_) on plant–pollinator interactions is poorly understood. This study provides the first systematic review of this topic and identifies important knowledge gaps. In addition, we present field data assessing the impact of eCO_2_ (150 ppm above ambient) on bluebell (*Hyacinthoides non-scripta*)–pollinator interactions within a mature, deciduous woodland system. Since 1956, only 71 primary papers have investigated eCO_2_ effects on flowering time, floral traits and pollination, with a mere 3 studies measuring the impact on pollination interactions. Our field experiment documented flowering phenology, flower visitation and seed production, as well as the abundance and phenology of dominant insect pollinators. We show that first and mid-point flowering occurred 6 days earlier under eCO_2_, but with no change in flowering duration. Syrphid flies and bumble bees were the dominant flower visitors, with peak activity recorded during mid- and late-flowering periods. Whilst no significant difference was recorded in total visitation or seed set between eCO_2_ and ambient treatments, there were clear patterns of earlier flowering under eCO_2_ accompanied by lower pollinator activity during this period. This has implications for potential loss of synchrony in pollination systems under future climate scenarios, with associated long-term impacts on abundance and diversity.

## 1. Introduction

Insect-mediated pollination is required by the majority of angiosperm species in order to achieve sexual reproduction [1]. This ecosystem function has a direct impact on plant reproduction and turnover in many ecosystems, including agricultural systems and, therefore, critically underpins our food security. Phenological synchrony with plant flowering times is crucial for many insect species utilising floral resources as a food source, as well as for organisms at higher trophic levels feeding on (or parasitising) these insect pollinators (e.g., [2]). There is now clear evidence that climate change is increasing global mean temperatures and changing patterns of precipitation, which has affected the phenology and physiology of plants and their pollinators [3,4,5,6]. This, in turn, has led to adverse impacts such as a phenological mismatch between plants and their pollinators [7,8]. Such impacts could result in realignment of interaction networks, changes in populations and may be a significant contributing factor to observed declines amongst many pollinator species, including local extinctions [9].

Elevated concentrations of atmospheric carbon dioxide (eCO_2_) are also hypothesised to influence plant–pollinator interactions, e.g., through impacts on plant growth, biochemistry, physiology, and phenology [10,11]. However, assessments of the potential impact of eCO_2_ on pollination interactions are limited. This is an important gap in our knowledge because any negative impacts eCO_2_ confers on these interactions could adversely impact populations of both the plants and their pollinators. This would represent yet another stressor potentially contributing to declining pollinator populations in combination with other factors such as habitat loss and fragmentation, agrochemicals, pathogens and alien species [12].

A key mechanism by which eCO_2_ is expected to impact pollination is through changes in plant flowering phenology. The term ‘flowering phenology’ comprises several constituent events including floral bud burst, maturation and release of fertile pollen, production of nectar, the stigma becoming receptive to pollen and floral senescence. A review by Springer and Ward [13] of studies across a range of cultivated and wild plants found varied responses of flowering time to eCO_2_. Light (photoperiod and illuminance), temperature (soil and air), nutrient availability (N, P, etc.) and water availability (precipitation and soil moisture) are also all known to affect flowering time [14,15], further complicating the picture. Many of the studies examining the direct effect of eCO_2_ on flowering phenology have used propagated plants in controlled environments, but this approach misses potential interacting effects of other variables such as local environmental/microclimate conditions and species interactions. Given that the scale and direction of the response to eCO_2_ can vary depending on species and context, further empirical studies are necessary in order to explore these responses in different species and systems, particularly in situ ecosystem (field)-scale experiments.

Another mechanism through which eCO_2_ could affect pollination is by altering the amount and/or biochemical composition of floral resources. Pollen and nectar are the primary currency in plant–pollinator interactions, and so any changes in the quantity or quality of this resource could have significant impacts on flower-visiting insects. Pollen is an important protein and lipid source for many insect species, including hoverflies (e.g., [16]), and is vital for obligate palynivores such as bees. Whilst there has been more focus on the impact of eCO_2_ on pollen rather than nectar, studies are still scarce. There is evidence that eCO_2_ leads to a decrease in pollen quantity in some horticultural species, such as *Lycopersicon lycopersicum* and *Cucurbita pepo* [17], and a decline in pollen quality (protein content) in *Solidago* spp. [18]. In other species, however, the reverse was noted, with increased pollen production under eCO_2_ in species such as ragweed, *Ambrosia artemisiifolia* [19] and Loblolly Pine, *Pinus taeda* [20]. Nectar can be a rich source of both amino acids and sugars [21]. There is evidence that the volume, sugar concentration, and sugar composition of nectar are all influenced by temperature and water availability [22], yet data on the direct effects of eCO_2_ on nectar production or composition are very limited. We can find only 10 studies assessing the effect of eCO_2_ on nectar, again with varying responses. For example, Lakes and Hughes [23] reported a reduced nectar volume, whilst López-Cubillos and Hughes [17] noted an increase in nectar production. eCO_2_ is known to increase C:N ratios and alter the nutritional value of plant tissue such as leaves [24,25], but it remains untested whether similar changes occur in pollen or nectar biochemistry. What is clear, is that changes in nutritional quality or quantity of floral resources can have significant consequences for pollinator development and reproductive success [26], as well as immune/disease responses and overall health [27]. Thus, examining the role of eCO_2_ within the context of pollinator nutritional ecology will be a key part of understanding plant–pollinator interactions, coevolution, and the restoration of declining pollinator populations under climate change.

Beside the impact on phenology and floral resources, it is also possible that eCO_2_ may affect pollination via other pathways, such as interfering with the production or detection of floral volatiles and thus disrupting plant–pollinator communication [28]. As pollination is a complex, multispecies, ecological interaction, it is inherently difficult to detect, disentangle and predict how it is impacted by shifting environmental factors such as eCO_2_. Empirical data from in situ, ecosystem-scale experiments are required, therefore, to answer these complex questions.

Exploring the consequences of eCO_2_ on ecosystem processes such as pollination is particularly difficult in complex ecosystems such as forests. This is due to the difficulty of manipulating CO_2_ concentrations at an appropriate spatial scale. Free Air Carbon Dioxide Enrichment (FACE) experiments are an invaluable tool in this context, where unenclosed forest/woodland plots are fumigated in situ and ecosystem responses measured to provide vital real-world data [29]. There are currently only two large-scale forest FACE experiments running globally. In the southern hemisphere, EucFACE (Australia) has been fumigating a eucalyptus forest with CO_2_ since 2012 [30], but this facility has yet to publish any studies on pollinator systems. In the northern hemisphere, the ‘Birmingham Institute of Forest Research’ (BIFoR) FACE facility (UK) has been fumigating a mature oak woodland system with CO_2_ since 2017, and provides the perfect opportunity to examine the impact of eCO_2_ on plant–pollinator interactions in this important temperate ecosystem.

Against this background, the current study had the following objectives: (1) to undertake the first systematic review of the literature investigating the effect of eCO_2_ on floral traits and pollination in order to highlight key knowledge gaps for future study, and (2) to assess the impact of eCO_2_ on plant–pollinator interactions within a mature, deciduous woodland, using the common bluebell (*Hyacinthoides non-scripta*) as a case study.

## 2. Materials and Methods

### 2.1. Systematic Review

A systematic review was preformed to provide a transparent, comprehensive and objective overview of the quantity and quality of evidence related to pollination under eCO_2_, following published guidelines [31]. A comprehensive search of the literature was performed in January 2020 and repeated in December 2020 using the online database Web of Knowledge (WoK version 5. 3), in English language only. A scoping process was performed to optimise the search terms so that the search was as comprehensive as possible whilst reducing the volume of irrelevant material. The final search terms used were: Title = ((CO_2_ OR “carbon dioxide”) AND ((‘flower* time’ OR ‘flower* phenology’) OR (pollinat* OR nectar OR pollen))). The search was also performed using the same search terms in the online search engine Google Scholar, and the first 80 results, sorted by relevance, were included. The results of the search were assessed against the inclusion criteria by examination of the abstract, and further exploration of the text where this was unclear. The inclusion criteria were set as: (1) the article must report the results of a primary empirical study, (2) the explanatory variables must include eCO_2_, and (3) the response variables must include either flowering phenology, floral resources or pollination. Studies reporting effects on reproductive allocation, fruit production or seed production were not included. Any article that did not pass all 3 inclusions criteria, or was a duplicate, was excluded.

Review papers included in search results, which passed all other inclusion criteria, were then further examined to identify any additional primary research articles. The final set of articles that had passed the inclusion criteria were read in full and entered into the database by extraction of the relevant data (Appendix A).

### 2.2. Field Experiment

#### 2.2.1. Location

The field experiment was conducted at the Birmingham Institute for Forest Research Free-Air CO_2_ Enrichment (‘BIFoR FACE’) experimental facility, located in Staffordshire, UK (52°47′58″ N, 2°18′15″ W) as described in Hart et al. [32]. The facility is located within a semi-natural, mature, temperate woodland with English oaks, *Quercus robur*, as the dominant tree species and an understory comprised mainly of common hazel, *Corylus avellana*. In brief, 3 experimental arrays fumigate 30 m diameter plots with 150 ppm above ambient CO_2_, with 3 control arrays which fumigate with ambient air. Fumigation commenced on 3 April 2017, which means the woodland system had been exposed to eCO_2_ for a period of 2 years prior to the experiment.

#### 2.2.2. Plant Study System

The common bluebell (*Hyacinthoides non-scripta*, Asparagaceae) is a widespread spring-flowering bulbous perennial that occurs throughout Atlantic Western Europe. It is an ideal case study species to explore how field-layer flowering plants within temperate woodlands might respond to eCO_2_ due to its abundance, floral composition, flowering phenology and insect-mediated pollination. The species is locally abundant throughout the experimental site in both eCO_2_ and ambient arrays. Typically, 7–20 flowers are produced on a raceme that opens in an acropetal sequence, each lasting 2–3 weeks. This species reproduces vegetatively by budding and sexually by seed. Insect-mediated cross-pollination is required to produce a full seed set, with self-pollinated flowers producing fewer fruits and seeds, conferring a degree of ‘effective self-incompatibility’ [33]. Each array contained a single patch of bluebells with a mean area of 3.5 m^2^ (SE = ±1.3) in the ambient arrays and 8.2 m^2^ (SE = ±3.8) in the treatment arrays (patch sizes ranged from 0.7 to 12.6 m^2^, Table 1).

#### 2.2.3. Environmental Data

To determine whether other environmental variables differed significantly among experimental arrays, soil temperature (°C), soil moisture (m^3^/m^3^) and patch-level illuminance (LUX) were recorded. Soil moisture was measured using CS655 probes (Campbell Scientific, Utah, UT, USA) and recorded on a Campbell Scientific CR300 series datalogger. Mean monthly soil moisture and temperature were calculated for the three years preceding this study. Mean daily soil moisture and temperature were also recorded throughout the duration of the flowering period. Illuminance was recorded for each patch throughout the flowering period using a smartphone light meter application (Lux Meter, My mobile tools dev, Android, Sony Xperia Z1 compact, Sony Europe, Weybridge, UK).

#### 2.2.4. Flowering Phenology

Trail cameras (SAS-DVRODR05, Konig, Edmonton, AB, Canada) were used to monitor the flowering phenology of the bluebells throughout the 2019 flowering period. Cameras were installed facing each experimental patch at a height of ~50 cm and took photographs twice a day. From these photographs, the date of specific flowering stages (first flowering date, mid-flowering date and flowering duration) was determined. Total flowers in bloom were counted weekly during in-person surveys.

#### 2.2.5. Insect Visitation Surveys

A 30 min flower visitation survey was conducted at each patch every two weeks throughout the bluebell flowering period (3 time points). The flowering period was subsequently divided into three time windows around these survey points (‘early’, ‘mid’ and ‘late’) to facilitate analysis. Surveys of all patches were performed in succession on the same day between 11:00 and 14:00, in a random order. Surveys were conducted on days when air temperature, wind, precipitation and cloud cover were as similar as possible. During each survey, every visit made by an insect to a bluebell flower within the patch was recorded and the insect identified to species level, or genus level for taxa where this is not possible from field identification (or family level for Ichneumonidae). A ‘visit’ was defined as each individual event when an insect entered/landed on a flower, potentially coming into contact with the floral reproductive organs (cf. [34]).

#### 2.2.6. Seed Counts

After all flowering was completed, 60 racemes were collected from each patch. The number of flowers produced and fruits that developed were recorded. The total number of seeds developing within 3 fruits from each raceme was then counted. One early-, one mid- and one late-flowering fruit were selected. This was determined by their position on the raceme, which corresponds to the period in which they flowered.

#### 2.2.7. Statistical Analyses

The impact of the treatment on first flowering date, mid-flowering date and flowering duration was assessed by ANOVA. Pearson’s product-moment correlations were performed between each of the flowering date measures (day of year first flowering, day of year mid-point flowering and flowering duration), and each environmental variable during the flowering period (light intensity, soil moisture and soil temperature). Comparisons of mean monthly soil moisture, soil temperature and mean light intensity during flowering period between treatment and ambient arrays were performed using Wilcoxon rank-sum tests.

The relationship between number of visits to a patch and patch size was tested with a linear regression. To analyse the effect of eCO_2_ treatment on number of visits per unit area of each patch, a Wilcoxon rank-sum test was used. The effect of time period on number of visits was tested using a Kruskal–Wallis rank sum test. The effects of both treatment and time period on seed set were tested by fitting generalised least squares model and applying the *varIdent* weights term to control for the heterogeneity in the sample period using nmle package version 3.1-144 [35]. The analysis of the interaction between mean number of seeds per fruit and mean number of flower visits was performed using a generalised linear model with Gaussian errors distribution. All statistical analyses were performed in R, version 3.5.2 (R Core Team, 2015). All the analyses were validated by examining model residuals (where appropriate) using model fits and inspection of model covariates residual spreads [36].

## 3. Results

### 3.1. Systematic Review

The search process returned a total of 189 articles—74 articles from Web of Knowledge, 80 from Google Scholar and 35 from examination of reference lists within review articles. Of these, 73 articles passed the inclusion criteria, with publication years ranging from 1956 to 2020 (Appendix A).

The mean treatment concentration of CO_2_ for these 73 studies was 730 ppm, with a mean control concentration of 360 ppm. More than 118 plant species from 32 families were investigated with 146 individual species level responses reported. There is a strong bias in the geographic location of the studies, with over 72% performed in North America (53%) or Europe (19%). There were two or fewer studies from Africa, Central America or the Middle East, and none from South America.

Fifty-five articles investigated the impact on eCO_2_ on flowering phenology (Figure 1, SM2). Flowering time varied from −60 to +10.8 days under eCO_2_ compared to controls, with a mean response of −3.73 days. The greatest mean advance in flowering date under eCO_2_ was exhibited by Ericaceae (−60), Solanaceae (−11.67) and Euphorbiaceae (−9), whereas the greatest mean delay in flowering was by Geraniaceae (+1.88), Amaranthaceae (+2.03) and Cucurbitaceae (+10.8).

Fifteen studies included eCO_2_ effects on pollen (Figure 1, Appendix A), with six reporting an increase in pollen production and four showing an associated decrease in quality through reduced protein content or increased metabolites. There were mixed results of the effect of eCO_2_ on nectar, with three articles showing increased production, two decreased and three with no change. Similarly, the response of nectar sugar content varied with two and one studies reporting increasing and decreasing concentration, respectively. Three studies directly measured the impact of eCO_2_ on more than just floral traits (Figure 1, Appendix A), of which two looked at a single crop species ex situ. These investigations found either increased visitation or decreased pollinator longevity, but neither were significant.

### 3.2. BIFOR FACE Field Experiment

#### 3.2.1. Environmental Parameters

Over the course of the proceeding three years mean soil temperature did not differ significantly between treatment and control arrays with an overall mean of 9.7 °C (±0.65) and 9.5 °C (±0.64), respectively (*p* = 0.7613, Appendix A). Mean soil moisture was also not significantly different over the same period with an overall mean of 16.27 m^3^/m^3^ (±1.02) in eCO_2_ arrays and 14.86 m^3^/m^3^ (±1.26) in ambient arrays (*p* = 0.3928, Appendix A). During the flowering period, mean light intensity in eCO_2_ arrays was 2852 lx (±314) and 3420 lx (±491) in ambient arrays (*p* = 0.361, not significant).

#### 3.2.2. Bluebell Flowering Phenology

Under eCO_2_ the mean date of first flower opening advanced by 6 days relative to the ambient control and the mean mid-point, between first flower opening and final flower senescing, also advanced by 6 days (Figure 2a, Appendix A). The advance of mid-flowering date under eCO_2_ was statistically significant (F = 12.893, *p* = 0.02295). The duration of flowering was not significantly different between treatment and control (F = 0.0091, *p* = 0.9286), with a mean of 46 days under eCO_2_ and 45 days for ambient patches (Appendix A). Mean peak flowering occurred in the late period for ambient patches, whereas mean peak flowering shifted to during the mid-flowering period in eCO_2_ patches (Figure 2b).

Pearson’s correlations of light intensity, soil temperature and soil moisture were non-significant between first flowering date (*p* = 0.1411, *p* = 0.4806, *p* = 0.1127), mid-flowering date (*p* = 0.2226, *p* = 0.6628, *p* = 0.2215) and flowering duration (*p* = 0.3167, *p* = 0.3255, *p* = 0.219) (Appendix A).

#### 3.2.3. Insect Visitation and Seed Production

Insect visitation of bluebells commenced as soon as the first flowers opened, but at low rates with a mean of 1 visit/patch during the early-flowering period (30 min observation periods, Appendix A). Visitation was significantly lower in the early-flowering period compared to the later flowering periods (*p* = 0.0436), with the mean number of visits per patch rising to 5.8 and 4.8 in the ‘mid’ and ‘late’ flowering periods, respectively (Figure 3a).

The overall number of visits under eCO_2_ were much higher than the overall number of visits under ambient CO_2_. However, visits were significantly correlated with patch size (*p* = 0.0029, R^2^ = 0.8914). There were no significant differences in visitation per area between treatment and control arrays (*p* = 0.6866).

A total of 18 species/species groups visited bluebell flowers during the surveys (Appendix A), of which 10 made repeated visits (Figure 3b). Hoverflies (Diptera: Syrphidae) were the most frequent visitor during all three flowering periods, contributing 55.7% of total visits (Figure 3a). The hoverflies *Platycheirus* spp. made the greatest number of visits, peaking in the mid-flowering period. *Rhingia campestris* made the second highest number of visits of any hoverfly species, with 88% of these occurring in the late flowering period. Bumble bees (Apidae: *Bombus*) represented 22.9% of total flower visits, with *B. pratorum* workers the most common bumble bee observed. For all other *Bombus* species, visits were made exclusively by queens.

Seed set followed a similar temporal pattern (Figure 4, Appendix A), with an initial mean of 4.91 seeds per fruit from early-flowering fruits. This increased to 7.48 mean seeds per fruit for mid-flowering period, and 6.30 mean seeds per fruit during the late-flowering period (*p* = 0.0526). There was a significant correlation between total number of flower visits recorded and mean number of seeds per fruit produced from flowers that bloomed during the corresponding period (*p* = 0.0348).

## 4. Discussion

### 4.1. Systematic Review

Compared with the number of studies that examine the effect of other climate change variables, such as temperature and precipitation patterns [12], there is a paucity of peer-reviewed literature that investigates the impact of eCO_2_ on pollination. The systematic review of the literature revealed here indicates that the majority of publications examining pollination under eCO_2_ reported on the impacts on flowering time (75%), which is likely due to this variable being easier to measure from direct observation. This covered a wide range of flowering species from a reasonable phylogenetic spread of families, although the importance of insect pollination to their pollination ecology varied considerably. For example, the large proportion of studies reporting the effect on Poaceae (10), which are largely wind-pollinated [37].

A key finding of this review is the net advance in flowering phenology under eCO_2_ by 3.73 days. This conforms with the general findings of previous studies assessing large numbers of species (e.g., [13]) and adds further evidence to the conclusions that increasing atmospheric CO_2_ will disrupt flowering phenology, leading to an advance in flowering for many species. This may be due to related increases in growth rates in response increasing photosynthetic rates [38]. The phylogenetic spread of phenological responses to eCO_2_ across various plant families is also potentially interesting. However, in this review, the strongest mean responses, in either direction, are underpinned by the findings of a smaller number of studies. For example, five of the six families with the strongest mean response, in either direction, are all derived from the results of a single study. Therefore, whilst the findings of this review suggest there is a general pattern in the response across a broad plant phylogeny, more studies would be needed to provide a robust assessment and with multiple assessments of individual plant families.

There were substantially fewer studies examining the impact of eCO_2_ on floral resources, which is likely due to the additional methodological steps required to sample and measure these properties. Where this was measured the results are equally mixed, i.e., no consistent response across species. Direct measurements of floral traits such as flowering time and floral resources do not provide a direct quantification of pollination, however, and therefore can only be used to infer the impact of eCO_2_ on this interaction.

Insect-mediated pollination is a complex interaction between multiple species, which is perhaps why so few studies to date have directly measured the impact of eCO_2_ upon it. Only 4% of the studies found by this review directly measured an aspect of the effect of eCO_2_ on pollination, revealing an important knowledge gap. Furthermore, none of these studies were performed in situ, therefore potentially missing the effects of important interactions that cannot be replicated in controlled environments. Empirical studies addressing this specific area are urgently needed to improve our predictions of plant–pollinator interaction changes under climate change.

### 4.2. BIFoR FACE Field Experiment

To our knowledge, this is the first study to empirically measure the impact of eCO_2_ on pollination directly in situ and the first assessment of a pollination interaction in a FACE experiment. Flowering phenology of common bluebell was found to advance by a mean of 6 days under eCO_2_ (Figure 2). This is consistent with the overall mean effects of eCO_2_ on flowering time reported in the articles included within the systematic review. Importantly, other variables such as soil moisture, soil temperature and light intensity, which are all known to influence phenology [39], did not vary significantly between eCO_2_ and ambient arrays within the BIFoR FACE facility for the 3 years preceding this study, meaning they are unlikely to help explain any observed differences in bluebell flowering between patches. This allows us to focus particular attention on the contribution of eCO_2_ on bluebell flowering traits, as well as plant–pollinator interactions. eCO_2_ is associated with increased growth in many plant species [40], which may lead to increased and/or earlier flower production. Whilst there was a greater number of flowers in the eCO_2_ arrays, this was directly related to patch size (Table 1), which was a pre-existing condition of the distribution of bluebell across the site. Neither flowers per area, nor flowers per raceme varied significantly between treatment and control arrays, therefore there is no evidence of an increased reproductive allocation under eCO_2_, although this cannot be fully assessed without further historical data on flower production per patch prior to fumigation.

The temporal patterns of insect visitation, and the associated consequences for seed set (Figure 3 and Figure 4), suggest that early flowers are less successful for *Hyacinthoides non-scripta.* Earlier flowering would, therefore, lead to decreased overall pollination success if the pollinator community did not also experience phenological advances at the same rate. Bluebell flowers are generally long-lived compared to many other woodland flowers [41], and are visited, and therefore pollinated, by a relatively generalist community of pollinators. The risk of phenological mismatch due to differential advances in phenology under eCO_2_ would potentially be greater for plants with shorter flowering periods and/or more specialist pollinators.

Many of the dominant pollinator species recorded visiting bluebells in this study, such as *Bombus pratorum*, *B. lapidarius*, *B. pascuorum*, *B. terrestris*, *Rhingia campestris* and *Platycheirus* spp., have been previously recorded visiting bluebells [42]. This adds to the evidence that they are the key species in the bluebell pollination interaction networks in Britain. Many of these hoverflies and bumble bees were also observed to be covered in bluebell pollen whilst moving between racemes and are, therefore, highly likely to be facilitating pollination. This is not true of all insect visitors, however, with numerous interactions failing to transfer pollen to a receptive stigma of a conspecific flower. Thus, visitation does not always equal pollination. In order to avoid conflation of ‘pollinators’ and ‘flower visitors’, studies that attempt to measure pollination must also measure pollination success, e.g., by measuring seed set. Corbet [33] found insect pollination to be directly related to seed set in bluebells. Our results support this, further suggesting that seed set may be a useful indicator of pollination success for this species.

Over 90% of all insect visits to bluebells occurred during the mid- and late-flowering periods, with many important (based on number of visits) species (e.g., *Rhingia*) only active in the late period. The advance in flowering phenology driven by eCO_2_ observed in this study, therefore, is potentially shifting peak flowering (and thus seed production) away from the peak flight period of key pollinators. Other insect species may of course ‘step in’ to provide a pollination service, but recent evidence from alpine environments suggests altering pollinator communities can have significant negative effects on plant reproductive success [43] and there is often less redundancy in pollination service provision for species that emerge early in the year. Negative impacts on important pollinator species could also be significant, for example it is worth noting that for many of the *Bombus* species observed, only queens were recorded visiting bluebells. Thus, this flower likely represents an important resource for queens emerging, somewhat nutrient-deprived, from their overwintering diapause. Any reduction in the availability of this resource, e.g., resulting from a phenological mismatch, could reduce the success of queens subsequently establishing a colony.

This study highlights the importance of in situ, field-scale experiments to test the impact of climate change variables, such as eCO_2_, on pollination. Several lessons may be gleaned from the data presented, which would be advantageous for future empirical studies seeking to further quantify the impact on eCO_2_ on insect flower visitation, and associated pollination success. Firstly, insect visitation to bluebells was relatively low, particularly early in the flowering period. Longer, synchronous observation periods, therefore, may be useful to provide data on a greater number of flower visits. Differences in visitation across patches may have been influenced by patch size, and therefore, relative apparentness and attractiveness to flower-visiting insects. Consistent patch sizes would control for much of this variation.

The phenological relationships between plants and insect pollinators are, of course, influenced by many factors other than eCO_2_, and while neither temperature nor precipitation had a significant effect in the current study, their influence across longer time scales is clearly evident. Both factors are known to affect flowering phenology [44], and temperature (importantly not just during spring) seems to be the dominant factor influencing insect emergence following winter diapause [4]. Indeed, global shifts in the synchrony of multiple species interactions, based on historic data, appear to be driven by temperature [45], but predictive models are now also needed in order to plan more effective conservation and food security strategies. The current study indicates that models for any plant–insect interactions would be wise to include eCO_2_ as a parameter in order to determine whether it will either exacerbate or reduce temperature-driven changes in phenological synchrony.

## 5. Conclusions

eCO_2_ is likely to directly impact plant–pollinator interactions in addition to other climate change variables, yet few studies have directly measured these impacts. Our results showed a consistent advancement of bluebell flowering under eCO_2_ in a deciduous woodland, which is also consistent with the mean effect established across studies of other plant species (in both lab and field settings). If this pattern continues under future eCO_2_ scenarios, then greater phenological mismatches may occur than predicted by temperature-based models alone, with the main flowering period of several plant species potentially losing synchrony with the peak flight period of key pollinators. This could lead to a shift in the plant–pollinator network, resulting in a declining forage resource for certain pollinators, as well as a decrease in in plant seed set. Importantly, this impact is likely to be greater for plant species with short flowering periods and/or very specialised plant–pollinator relationships.

## Figures and Tables

**Figure 1 insects-12-00512-f001:**
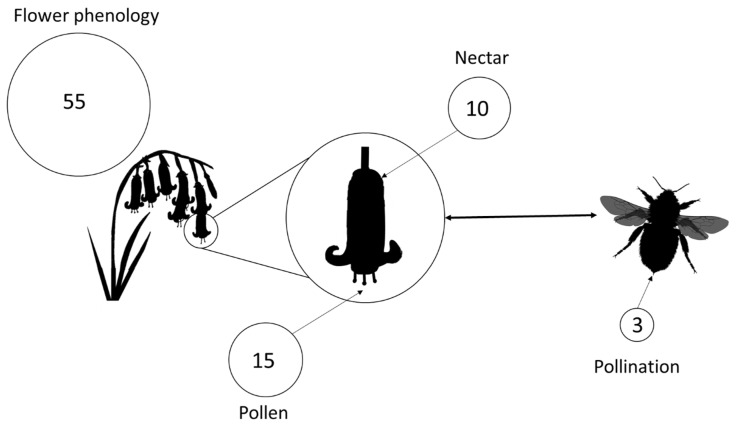
Components of common bluebell, *Hyacinthoides non-scripta*, pollination interaction and the number of published studies (on any species) considering the impact of eCO_2_ on each component. The area of the circle is proportional to the number of studies.

**Figure 2 insects-12-00512-f002:**
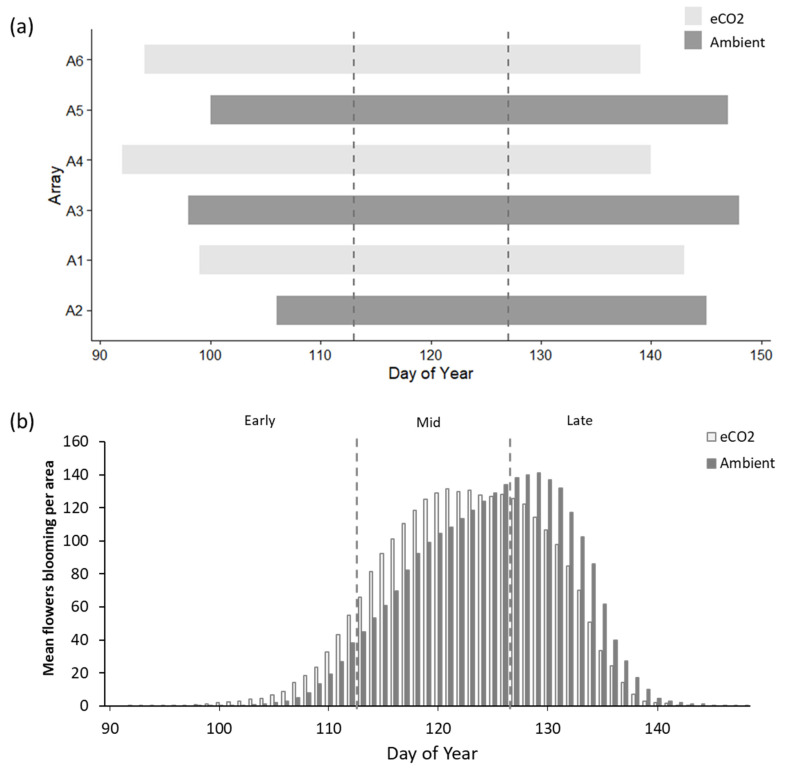
(**a**) Patch-wide flowering period of the common bluebell, *Hyacinthoides non-scripta*, per FACE array. Bars commence on the day of year when the first flower opened and end when last flower senesced. Dashed lines denote the boundaries for ‘early’, ‘mid’ and ‘late’ time windows within flowering period. (**b**) Daily mean total number of flowers blooming per patch area (m^2^) for eCO_2_ and ambient patches. Totals were based on weekly counts and interpolation for missing values, cross-referenced against daily phenology photographs.

**Figure 3 insects-12-00512-f003:**
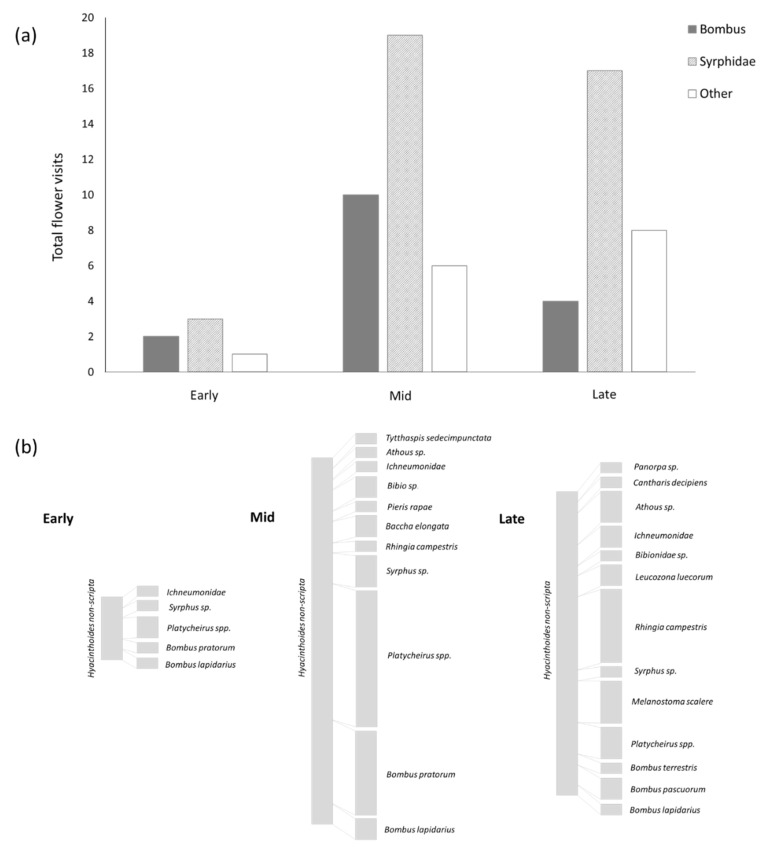
(**a**) Total number of flower visits made by bumble bees (*Bombus*), hoverflies (Syrphidae) and all other flower visitors during each time period. Data are based on 30 min observations at each array for each time period during flowering. (**b**) Sankey diagram of the visitation network during each flowering time period. Bar size is proportional to total number of visits by each taxon during each time period.

**Figure 4 insects-12-00512-f004:**
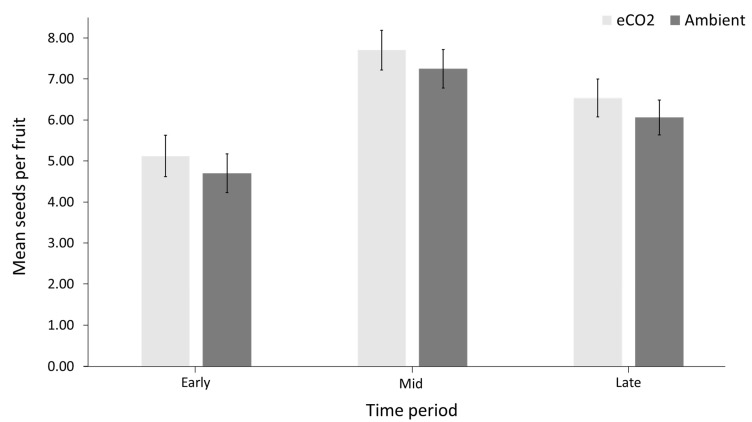
Mean seeds per fruit for eCO_2_ vs. ambient for each time period during flowering. *n* = 60 fruits per patch (total 360) at each time point. Error bars denote ± standard error of the mean.

**Table 1 insects-12-00512-t001:** Bluebell patch metrics for each experimental array.

Array	Treatment	Patch Size (m^2^)	Total Number of Racemes	Total Number of Flowers	Total Number of Fruits
1	eCO_2_	0.71	60	257	167
2	Ambient	1.60	88	581	405
3	Ambient	6.03	150	933	613
4	eCO_2_	11.31	230	1495	985
5	Ambient	2.90	96	536	285
6	eCO_2_	12.57	250	1879	1279

## Data Availability

The full dataset, including the systematic review data matrix, can be found in the supporting materials.

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
