# Peer review of "Elevated CO_2_ Impacts on Plant–Pollinator Interactions: A Systematic Review and Free Air Carbon Enrichment Field Study"

_insects, 2021, doi:10.3390/insects12060512_

Round 1

Reviewer 1 Report

An interesting study, worth publishing, but the manuscript has some problems. There are a lot of editorial problems, which should have been fixed by the authors before submission, as they're distracting to reviewers.  I've made a lot of corrections/suggestions on the PDF.  

I think that some relevant literature has not been cited, and included a few examples. 

The figures and tables are clear and useful.  I think the conclusions are warranted. 

Reviewer 2 Report

This manuscript is well written and successfully combines a literature review with an experimental field study. That said in the introduction it could already be mentioned why bluebell pollination is a model system as opposed to just a case study. The manuscript can make stronger recommendations on how to quantify ECO2 impacts on insect pollination of flowering plants. For example site selection and sampling duration.

Specific comments:

lines 43-46: These lines refer to an ecosystem service while in fact the way it is stated it is an ecosystem function, as it talk about angiosperm pollination in a general sense. Later when you mention food security and imply crop pollination, can you then only use the term ecosystem service.

lines 129 and 284: species name not in italics.

Figure 2a: The last two bars (Array 1 and 2) should be swapped round to form a consistent pattern of eCO2 vs. ambient.

Lines 174-176: Why were bluebell patches exposed to ambient conditions smaller than those in the experimental arrays? Surely you had more options to select the control patches as the experimental area is supposed to the limiting factor. Can you provide a reason for this?

Line 293 Insect visitation: summary of number of visits should be provided so that the reader can immediately see the amount of pollinator visits counted. 

Line 313: where you refer to total number of visits it is clear that very few insects flower visitors were counted. Is this not an indication that the 30 minutes of observation per patch was not sufficient? In fact, giver your studies objectives, your experiment should have compared the number of pollinators observed in the ambient versus the eCO2 patches. It appears that by not observing for enough time and having patch size vary strongly between treatment and control, has prevented this being tested.

Lines 386-391: The statement made is confusing. If early flowers flower for longer, does this not compensate for the effect of investing in early flowering when pollinator visitation is lower? I.e., if flowers are long lived and have low visitation rates at a given time, is this not compensated by flowering for longer? This section needs to be more thoroughly covered.

Line 415: How do you know that these were queen bumble bees?

End of report.

Round 2

Reviewer 1 Report

Authors have done a good job of responding to reviews of the previous version. I have made a few comments on the PDF, which should not take long to address. 

David Inouye

Author Response

Thank you for your additional comments. Please find below our responses to each point raised:

Line 241 & 306 – ‘SM’ refers to ‘supplementary material’. A summary of all supplementary material (SM1-SM8) can be found in the ‘supplementary material’ section on line 462.

Line 382 – A comma has been added.

Line 478 – Acknowledgement of the reviewers' helpful comments has been added. Thank you!